# RainNet: A Large-Scale Imagery Dataset and Benchmark for Spatial Precipitation Downscaling

**Xuanhong Chen**[1*]    **Kairui Feng**[2*]    **Naiyuan Liu**[3]    **Bingbing Ni**[1†]    **Yifan Lu**[1]
**Zhengyan Tong**[1]    **Ziang Liu**[1‡]

[1]Shanghai Jiao Tong University    [2]Princeton University    [3]University of Technology Sydney

{chen19910528, yifan_lu, 418004}@sjtu.edu.cn
{kairuif}@princeton.edu
{naiyuan.liu}@student.uts.edu.au
{ziang_liu}@brown.edu

## Abstract

AI-for-science approaches have been applied to solve scientific problems (e.g., nuclear fusion, ecology, genomics, meteorology) and have achieved highly promising results. Spatial precipitation downscaling is one of the most important meteorological problem and urgently requires the participation of AI. However, the lack of a well-organized and annotated large-scale dataset hinders the training and verification of more effective and advancing deep-learning models for precipitation downscaling. To alleviate these obstacles, we present the first large-scale spatial precipitation downscaling dataset named *RainNet*, which contains more than $62,400$ pairs of high-quality low/high-resolution precipitation maps for over 17 years, ready to help the evolution of deep learning models in precipitation downscaling. Specifically, the precipitation maps carefully collected in RainNet cover various meteorological phenomena (e.g., hurricane, squall), which is of great help to improve the model generalization ability. In addition, the map pairs in RainNet are organized in the form of image sequences (720 maps per month or 1 map/hour), showing complex physical properties, e.g., temporal misalignment, temporal sparse, and fluid properties. Furthermore, two deep-learning-oriented metrics are specifically introduced to evaluate or verify the comprehensive performance of the trained model (e.g., prediction maps reconstruction accuracy). To illustrate the applications of RainNet, 14 state-of-the-art models, including deep models and traditional approaches, are evaluated. To fully explore potential downscaling solutions, we propose an implicit physical estimation benchmark framework to learn the above characteristics. Extensive experiments demonstrate the value of RainNet in training and evaluating downscaling models. Our dataset is available at https://neuralchen.github.io/RainNet/.

## 1   Introduction

Deep learning has made enormous breakthroughs in the natural sciences (e.g., nuclear fusion [11], ecology [52], genomics [28], meteorology [3]), which is extremely good at extracting valuable knowledge from numerous amounts of data. In recent years, with computer science development, a deluge of earth system data is continuously being obtained, coming from sensors all over the

---

*Equal contribution.

†Bingbing Ni is the corresponding author.

‡Work done during undergraduate student at Shanghai Jiao Tong University.

36th Conference on Neural Information Processing Systems (NeurIPS 2022).

earth and even in space. These ever-increasing massive amounts of data with different sources and structures challenge the geoscience community, which lacks practical approaches to understand and further utilize the raw data [44]. Specifically, several preliminary works [16, 59, 21, 43, 1, 55] try to introduce deep-learning frameworks to solve meteorological problems, e.g., spatial precipitation downscaling.

In this paper, we focus on the spatial precipitation downscaling task [60]. Spatial precipitation downscaling is a procedure to infer high-resolution meteorological information from low-resolution variables, which is one of the most important upstream components for meteorological task [3]. The precision of weather and climate prediction is highly dependent on the resolution and reliability of the initial environmental input variables, and spatial precipitation downscaling is the most promising solution. The improvement of the weather/climate forecast and geo-data quality saves tremendous money and lives; with the fiscal year 2021 budget of over $1 billion, NSF funds thousands of colleges in the U.S. to research on these topics [38].

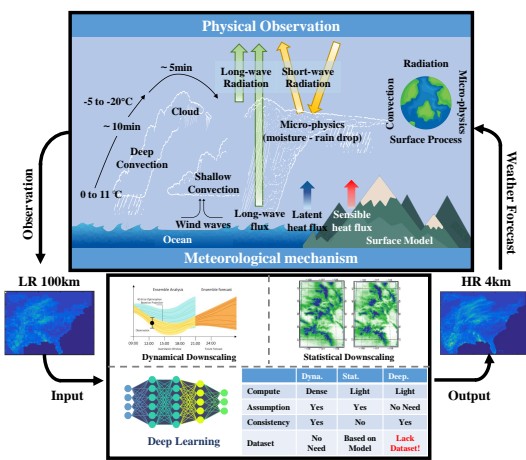

Figure 1: Meteorological physical process. Deep learning framework is one potentially promising solution, comparing to computational dense dynamic methods and spatial-temporal non-consistent statistical methods. No formal dataset challenges the application of deep learning in downscaling. In contrast to simulated toy-dataset, our RainNet contains more than 62, 400 low- and corresponding high-resolution precipitation map pairs.

Unfortunately, there are looming issues that hinder the research of spatial precipitation downscaling in the deep learning community: 1) Lack of "deep-learning ready" datasets. The existing deep-learning based downscaling methods are only applied to ideal retrospective problems and verified on simulated datasets (e.g., mapping bicubic of precipitation generated by weather forecast model to original data [4]), which significantly weakens the credibility of the feasibility, practicability, and effectiveness of the methods. It is worth mentioning that the data obtained by the simulated degradation methods (e.g., bicubic) is completely different from the real data usually collected by two measurement systems (e.g., satellite and radar) with different precision. The lack of a well-organized and annotated large-scale dataset hinders the training and verification of more effective and complex deep-learning models for precipitation downscaling. 2) Lack of tailored metrics to evaluate deep-learning based frameworks. Unlike deep learning (DL) communities, scientists in meteorology usually employ maps/charts to assess downscaling models case by case based on domain knowledge [21, 56], which hinders the application of RainNet in DL communities. For example, He et. al. [21] uses log-semivariance (spatial metrics for local precipitation) and quantile-quantile maps to analyze the maps. 3) The efficient deep-learning framework for downscaling should be established. Contrary to image data, the proposed real precipitation dataset covers various types of real meteorological phenomena (e.g., hurricane, squall), and shows the physical characters (e.g., *temporal misalignment*, *temporal sparse* and *fluid properties*) that challenge the downscaling algorithms. Traditional computationally dense physics-driven downscaling methods are powerless to handle the increasing meteorological data size and are flexible to multiple data sources.

To alleviate these obstacles, we propose the first large-scale spatial precipitation downscaling dataset named *RainNet*, which contains more than 62, 400 pairs of high-quality low/high-resolution precipitation maps for over 17 years, ready to help the evolution of deep models in spatial precipitation downscaling. The proposed dataset covers more than 9 million square kilometers of land area, which contains both wet and dry seasons and diverse meteorological phenomena. To facilitate DL/ML and other researchers to use RainNet, we introduce 6 most concerning indices to evaluate downscaling models: mesoscale peak precipitation error (MPPE), heavy rain region error (HRRE), cumulative precipitation mean square error (CPMSE), cluster mean distance (CMD), heavy rain transition speed (HRTS) and average miss moving degree (AMMD). In order to further simplify the application of indices, we abstract them into two weighted and summed metrics: Precipitation Error Measure (PEM) and Precipitation Dynamics Error Measure (PDEM). Unlike video super-resolution, the motion of

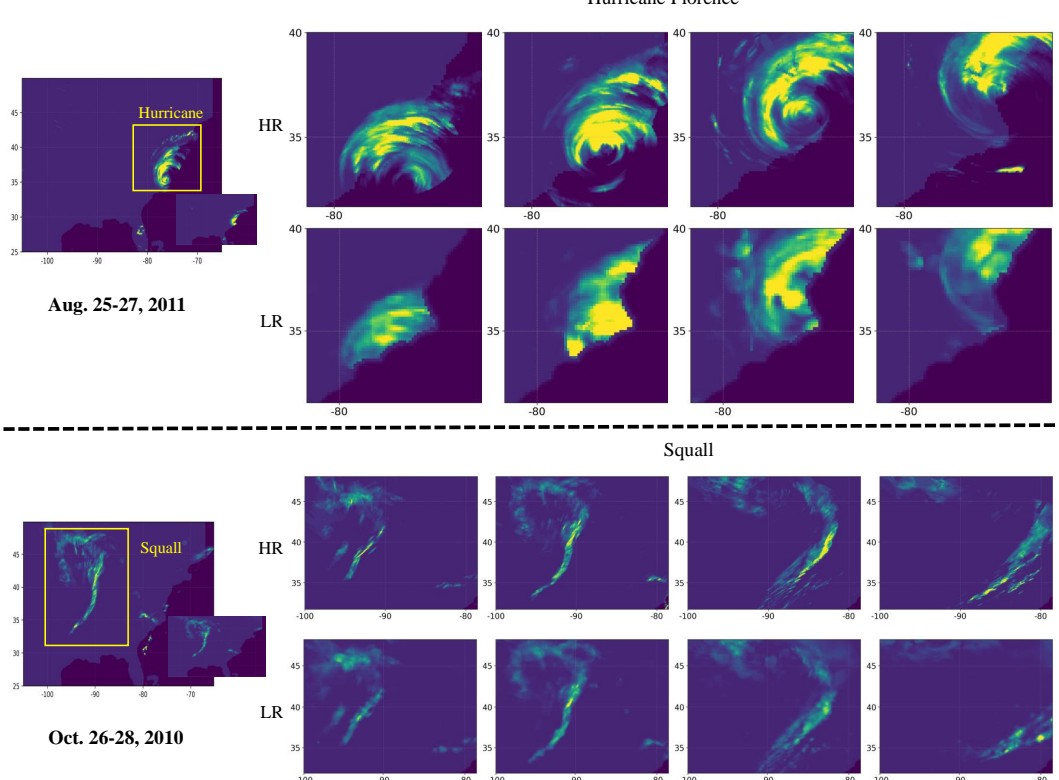

Figure 2: **Dataset Visualization**. Please zoom in the figure for better observation. Please note that the details of the precipitation map are partially lost due to file compression. Here we plot 2 groups of typical meteorological phenomena (hurricane and squall) in the dataset. To learn more about the dataset, please visit our project website and supplementary material. We show some video data of the dataset in the supplementary material.

the precipitation region is non-rigid (i.e., fluid), while video super-resolution mainly concerns rigid body motion estimation. To fully explore how to alleviate the mentioned predicament, we propose an implicit dynamics estimation driven downscaling benchmark model. Our model hierarchically aligns adjacent precipitation maps, that is, implicit motion estimation, which is very simple but exhibits highly competitive performance. Based on meteorological science, we also proved that the dataset we constructed contained the full information people may need to recover the higher resolution observations from lower resolution ones. We believe RainNet provides an imagenet-like track for natural science research in the deep-learning community. The main contributions of this paper are:

- To the best of our knowledge, we present the first REAL (non-simulated) large-scale spatial precipitation downscaling dataset for deep learning;

- We introduce 2 easy-to-use metrics to evaluate the performance of downscaling models;

- We propose a benchmark downscaling model with strong competitiveness. We evaluate 14 competitive potential solutions on the proposed dataset and analyze the feasibility and effectiveness of these solutions.

## 2  Background

At the beginning of the $19^{th}$ century, geoscientists recognized that predicting the state of the atmosphere could be treated as an initial value problem of mathematical physics, wherein future weather is determined by integrating the governing partial differential equations, starting from the observed current weather. Today, this paradigm translates into solving a system of nonlinear differential equations at about half a billion points per time step and accounting for dynamic, thermodynamic,

radiative, and chemical processes working on scales from hundreds of meters to thousands of kilometers and from seconds to weeks [3]. The Navier–Stokes and mass continuity equations (including the effect of the Earth's rotation), together with the first law of thermodynamics and the ideal gas law, represent the full set of prognostic equations in the atmosphere, describing the change in space and time of wind, pressure, density and temperature are described (formulas given in supplementary) [3]. These equations have to be solved numerically using spatial and temporal discretization because of the mathematical intractability of obtaining analytical solutions, and this approximation creates a distinction between so-called resolved and unresolved scales of motion.

## 2.1 Spatial Downscaling of Precipitation

The global weather forecast model is treated as a computational problem, and relying on high-quality initial data input. The error of weather forecast would increase exponentially over time from this initial error of the input dataset. Downscaling is one of the most important approaches to improving the initial input quality. Precipitation is one of the essential atmospheric variables that are related to daily life. It could easily be observed, by all means, e.g., gauge station, radar, and satellites. Applying downscaling methods to precipitation and creating high-resolution rainfall is far more meaningful than deriving other variables, while it is the most proper initial task to test deep learning's power in geo-science. The traditional downscaling methods can be separated into dynamic and statistical downscaling.

Dynamic downscaling treats the downscaling as an optimization problem constraint on the physical laws. The dynamic downscaling methods find the most likely precipitation over space and time under the pre-defined physical law. It usually takes over 6 hours to downscale a 6-hour precipitation scenario globally on supercomputers [10]. As the dynamic downscaling relies on pre-defined known macroscopic physics, a more flexible weather downscaling framework that could easily blend different sources of observations and show the ability to describe more complex physical phenomena on different scales is desperately in need.

Statistical downscaling is trying to speed up the dynamic downscaling process. The input of statistical downscaling is usually dynamic model results or two different observation datasets on different scales. However, due to the quality of statistical downscaling results, people rarely apply statistical downscaling to weather forecasts. These methods are currently applied in the tasks not requiring high data quality but more qualitative understanding, e.g., climate projection, which forecasts the weather for hundreds of years on coarse grids and uses statistical downscaling to get detailed knowledge of medium-scale future climate system.

## 3 RainNet: Spatial Precipitation Downscaling Imagery Dataset

### 3.1 Data Collection and Processing

To build up a standard *realistic (non-simulated)* downscaling dataset for computer vision, we selected the eastern coast of the United States, which covers a large region (7 million $km^2$; $105° \sim 65°W$, $25° \sim 50°N$, GNU Free Documentation License 1.2) and has a 20-year high-quality precipitation observations. It is worth mentioning that this region is selected for two reasons: on the one hand, this region has both LR and HR instrument observations, and on the other hand, it covers the various meteorological phenomena. We collected two precipitation data sources from National Stage IV QPE Product (StageIV [37]; high resolution at $0.04°$ (approximately $4km$), GNU Free Documentation License 1.2) and North American Land Data Assimilation System (NLDAS [62]; low resolution at $0.125°$ (approximately $13km$)). StageIV is mosaicked into a national product at National Centers for Environmental Prediction (NCEP), from the regional hourly/6-hourly multi-sensor (radar+gauges) precipitation analyses (MPEs) produced by the 12 River Forecast Centers over the continental United States with some manual quality control done at the River Forecast Centers (RFCs). NLDAS is constructed quality-controlled, spatially-and-temporally consistent datasets from the gauges and remote sensors to support modeling activities. Both products are hourly updated and both available from 2002 to the current age.

In our dataset, we further selected the eastern coast region for rain season ($July \sim November$, covering hurricane season; hurricanes pour over $10\%$ annual rainfall in less than 10 days). We matched the coordinate system to the lat-lon system for both products and further labeled all the

hurricane periods happening in the last 17 years. These heavy rain events are the largest challenge for weather forecasting and downscaling products. As heavy rain could stimulate a wide-spreading flood, which threatens local lives and arouses public evacuation. If people underestimate the rainfall, a potential flood would be underrated; while over-estimating the rainfall would lead to unnecessary evacuation orders and flood protection, which is also costly.

## 3.2 Dataset Statistics

At the time of this work, we collected and processed precipitation data for the rainy season for 17 years from 2002 to 2018. One precipitation map pair per hour, $24$ precipitation map pairs per day. In detail, we have collected $85$ months or $62424$ hours, totaling $62424$ pairs of high-resolution and low-resolution precipitation maps. The size of the high-resolution precipitation map is $624 \times 999$, and the size of the low-resolution is $208 \times 333$. Various meteorological phenomena and precipitation conditions (e.g., hurricanes, squall lines.) are covered in these data. The precipitation map pairs in RainNet are stored in HDF5 files that make up 360 GB of disk space. We select 2 typical meteorological phenomena and visualize them in Fig. 2. Our data is collected from satellites, radars, gauge stations, etc., which covers the inherent working characteristics of different meteorological measurement systems. Compared with traditional methods that generate data with different resolutions through physical model simulation, our dataset is of great help for deep models to learn real meteorological laws. The proposed dataset can be used under the Creative Common License (Attribution CC BY).

## 3.3 Dataset Analysis

In order to help design a more appropriate and effective precipitation downscaling model, we have explored the property of the dataset in depth. As mentioned above, our dataset is collected from multiple sensor sources (e.g., satellite, weather radar), which makes the data show a certain extent of *misalignment*. Our efforts here are not able to vanquish the misalignment. This is an intrinsic problem brought about by the fusion of multi-sensor meteorological data. Limited by observation methods (e.g., satellites can only collect data when they fly over the observation area), meteorological data is usually *temporal sparse*, e.g., in our dataset, the sampling interval between two precipitation maps is one hour. The temporal sparse leads to serious difficulties in the utilization of precipitation sequences. Additionally, the movement of the precipitation position is directly related to the cloud. It is a fluid movement process that is completely different from the rigid body movement concerned in image/video super-resolution. At the same time, the cloud will grow or dissipate in the process of flowing and even form new clouds, which further complicates the process. In the nutshell, although existed SR is a potential solution for downscaling, there is a big difference between the two. Especially, the three characteristics of downscaling mentioned above: *temporal misalignment*, *temporal sparse*, *fluid properties*, which make the dynamic estimation of precipitation more challenging.

# 4 Evaluation Metrics

Due to the difference between downscaling and traditional image/video super-resolution, the metrics that work well under SR tasks may not be sufficient for precipitation downscaling. By gathering the metrics from the meteorologic literature (the literature includes are [66, 35, 14, 21, 40, 61], we select 6 most common metrics (a metric may have multiple names in different literature) to reflect the downscaling quality: mesoscale peak precipitation error (MPPE), cumulative precipitation mean square error (CPMSE), heavy rain region error (HRRE) , cluster mean distance (CMD), heavy rain transition speed (HRTS) and average miss moving degree (AMMD). These 6 metrics can be separated as reconstruction metrics: MPPE, HRRE, CPMSE, AMMD, and dynamic metrics: HRTS and CMD.

The MPPE ($mm/hour$) is calculated as the difference of top quantile between the generated/real rainfall dataset which considers both spatial and temporal properties of mesoscale meteorological systems, e.g., hurricane, squall. This metric is used in most of these papers (for example [66, 35, 14, 21, 40, 61] suggest the quantile analysis to evaluate the downscaling quality).

The CPMSE ($mm^2/hour^2$) measures the cumulative rainfall difference on each pixel over the time-axis of the test set, which shows the spatial reconstruction property. Similar metrics are used

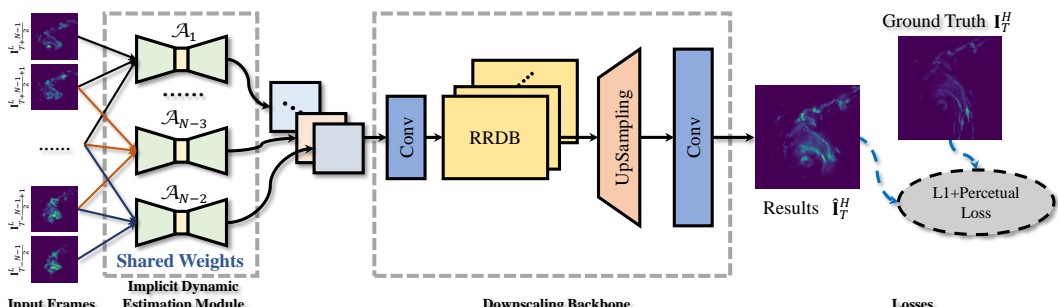

Figure 3: The pipeline of our proposed baseline model for spatial precipitation downscaling.

in [66, 35, 61] calculated as the pixel level difference of monthly rainfall and used in [21] as a pixel level difference of cumulative rainfall with different lengths of the record.

The HRRE ($km^2$) measures the difference in heavy rain coverage on each time slide between generated and labeled test set, which shows the temporal reconstruction ability of the models. The AMMD ($radian$) measures the average angle difference between main rainfall clusters. Similar metrics are used in [66, 35, 61] as rainfall coverage of a indefinite number precipitation level and used in [21, 40] as a continuous spatial analysis.

As a single variable dataset, it is hard to evaluate the ability of different models to capture the precipitation dynamics when temporal information is not included (a multi-variable dataset may have wind speed, a typical variable representing dynamics, included). So here we introduce the first-order temporal and spatial variables to evaluate the dynamical property of downscaling results. Similar approaches are suggested in [35, 14, 40]. The CMD ($km$) physically compares the location difference of the main rainfall systems between the generated and labeled test set, which could be also understood as the RMSE of the first-order derivative of precipitation data in spatial directions. The HRTS ($km/hour$) measures the difference between the main rainfall system moving speed between the generated and labeled test set which shows the ability of models to capture the dynamic property, which could be also understood as the RMSE of the first-order derivative of precipitation data on the temporal direction. Similar metrics are suggested in [35, 14, 40] as the auto-regression analysis and the differential analysis.

More details about the metrics and their equations are given in supplementary materials. One metrics group (MPPE, HRRE, CPMSE, AMMD) mainly measures the rainfall deviation between the generated precipitation maps and GT. The other group (HRTS and CMD) mainly measures the dynamic deviation of generated precipitation maps. So many measures will create difficulties in application and understanding. In order to further simplify the application of indices, we finally abstract them into two weighted and summed metrics: Precipitation Error Measure (PEM) and Precipitation Dynamics Error Measure (PDEM). We first align the dimensions of these two groups of metrics respectively. The first group of metrics (MPPE, HRRE, CPMSE, AMMD) is normalized, weighted , and summed to get the precipitation error measure (PEM). According to [17], all the metrics are transferred to Percent Bias (PBIAS) to be suitable for metrics weighting. The original definition of PBIAS is the bias divided by observation, as $PBIAS = |Q_{model} - Q_{obs}|/|Q_{obs}|$. Here we rewrite the original metrics to PBIAS by dividing the metrics with annual mean observations of the original variables (AMO), as $PBIAS_i^{PEM} = |Metrics_i^{PEM}|/|AMO_i^{PEM}|$, $Metrics_i^{PEM} = \{MPPE, HRRE, CPMSE, AMMD\}$. In our dataset, $AMO_{MPPE}^{PEM} = 64$, $AMO_{HRRE}^{PEM} = 533$, $AMO_{AMMD}^{PEM} = 0.64$, $AMO_{CPMSE}^{PEM} = 332$, $AMO_{HRTS}^{PEM} = 15$, $AMO_{CMD}^{PEM} = 26$. The metrics then are ensembled to a single metric (PEM) with equal weight, as $PEM = \sum_i 0.25 \cdot PBIAS_i^{PEM}$. Following the same procedure, we then ensemble the second group of dynamic metrics (HRTS and CMD) to a single metrics $PDEM = \sum_i 0.5 \cdot PBIAS_i^{PDEM}$. We also include the most commonly used metric RMSE as one single metric in our metrics list. RMSE could evaluate both the reconstruction and dynamic property of the downscaling results.

## 5    Benchmark of RainNet

As a potential solution, *Super-Resolution (SR)* frameworks are generally divided into the Single-Image Super-Resolution (SISR) and the Video Super-Resolution (VSR). Video Super-Resolution

is able to leverage multi-frame information to restore images, which better matches the nature of downscaling. We will demonstrate this judgment in Sec. 6.1. The VSR pipeline usually contains three components: deblurring, inter-frame alignment, and super-resolution. Deblurring and inter-frame alignment are implemented by the motion estimation module. There are four motion estimation frameworks: 1) RNN based [29, 49, 25, 19]; 2) Optical Flow [65]; 3) Deformable Convolution based [51, 63, 57]; 4) Temporal Concatenation [26, 6, 32]. In fact, there is another motion estimation scheme proposed for the first time in the noise reduction task [50], which achieves an excellent video noise reduction performance. Inspired by [50], we design an implicit dynamics estimation model for the spatial precipitation downscaling. It is worth mentioning that our proposed model and the above four frameworks together form a relatively complete candidate set of dynamic estimation solutions.

**Proposed Framework.** As shown in Fig. 3, our framework consists of two components: *Implicit dynamic estimation module* and *downscaling Backbone*. These two parts are trained jointly. Suppose there are $N$ adjacent low-resolution precipitation maps $\{\mathbf{I}^L_{T-\frac{N-1}{2}}, .., \mathbf{I}^L_T, ..., \mathbf{I}^L_{T+\frac{N-1}{2}}\}$. Here, we employ superscript $\cdot^H$ denotes the high resolution maps and $\cdot^L$ for the low resolution counterpart. The task is to reconstruct the high-resolution precipitation map $\mathbf{I}^H_T$ of $\mathbf{I}^L_T$. The implicit dynamic estimation module is composed of multiple vanilla networks $\mathcal{A} = \{\mathcal{A}_1, ..., \mathcal{A}_{N-2}\}$ ($N = 5$ in this paper) sharing weights. Each vanilla network receives three adjacent frames as input, outputs, and intermediate results. The intermediate result can be considered as a frame with implicit dynamic alignment. We concatenate all the intermediate frames as the input of the next module. The specific structure of the vanilla network can be found in the supplementary materials. The main task of the downscaling backbone is to restore the high-resolution precipitation map $\mathbf{I}^H_T$ based on the aligned intermediate frames. In order to make full use of multi-scale information, we use multiple Residual-in-Residual Dense Blocks [58] in the network. We employ the interpolation+convolution [39] as the up-sampling operator to reduce the checkerboard artifacts. After processing by downscaling backbone we get the final estimated HR map $\hat{\mathbf{I}}^H_T$.

**Model objective.** The downscaling task is essentially to restore high-resolution precipitation maps. We learn from the super-resolution task and also apply $\mathcal{L}1$ and perceptual loss [27] as the training loss of our model. The model objective is shown below:

$$\mathcal{L}(\hat{\mathbf{I}}^H_T, \mathbf{I}^H_T) = \parallel \hat{\mathbf{I}}^H_T - \mathbf{I}^H_T \parallel_1 + \lambda \parallel \phi(\hat{\mathbf{I}}^H_T) - \phi(\mathbf{I}^H_T) \parallel_2, \tag{1}$$

where $\phi$ denotes the pre-trained VGG19 network [47], we select the $Relu5 - 4$ (without the activator [58]) as the output layer. $\lambda$ is the coefficient to balance the loss terms. In our framework, we set $\lambda = 20$ to balance the order of magnitude between losses.

## 6 Experimental Evaluation

We conduct spatial precipitation downscaling experiments to illustrate the application of our proposed RainNet and evaluate the effectiveness of the benchmark downscaling frameworks. Following the mainstream evaluation protocol of DL/ML communities, cross-validation is employed. In detail, we divide the dataset into 17 parts (2002.7~2002.11, 2003.7~2003.11, 2004.7~2004.11, 2005.7~2005.11, 2006.7~2006.11, 2007.7~2007.11, 2008.7~2008.11, 2009.7~2009.11, 2010.7~2010.11, 2011.7~2011.11, 2012.7~2012.11, 2013.7~2013.11, 2014.7~2014.11, 2015.7~2015.11, 2016.7~2016.11, 2017.7~2017.11, 2018.7~2018.11) by year, and sequentially employ each year as the test set and the remaining 16 years as the training set, that is, 17-fold cross-validation. All models maintain the same training settings and hyperparameters during the training phase. These data cover various complicated precipitation situations such as hurricanes, squall lines, different levels of rain, and sunny days. It is sufficient to select the rainy season of the year as the test set from the perspective of meteorology, as the climate of one area is normally stable.

### 6.1 Baselines

The SISR/VSR and the spatial precipitation downscaling are similar to some extent, so we argue that the SR models can be applied to the task as the benchmark models. The input of SISR is a single image, and the model infers a high-resolution image from it. Its main focus is to generate high-quality texture details to achieve pleasing visual effects. In contrast, VSR models input multiple frames of

| Approach | MPPE↓ | HRRE↓ | AMMD↓ | CPMSE↓ | HRTS↓ | CMD↓ | PEM↓ | PDEM↓ | RMSE×100↓ |
|---|---|---|---|---|---|---|---|---|---|
| Kriging | 4.036 | 339.641 | 0.204 | 4.891 | 9.958 | 12.277 | 0.259 | 0.568 | 0.372 |
| Bicubic | 4.600 | 306.996 | 0.208 | 3.678 | 10.453 | 12.389 | 0.247 | 0.587 | 0.345 |
| SRCNN | 5.333 | 296.950 | 0.225 | 3.929 | 10.091 | 12.396 | 0.252 | 0.575 | 0.405 |
| SRGAN | 14.125 | 298.290 | 0.221 | 91.464 | 9.429 | 11.891 | 0.352 | 0.543 | 0.603 |
| EDSR | 4.748 | 288.354 | 0.204 | 3.292 | 9.605 | 12.259 | 0.236 | 0.556 | 0.329 |
| ESRGAN | 6.205 | 407.848 | 0.219 | 4.483 | 10.201 | 17.035 | 0.305 | 0.668 | 0.563 |
| DBPN | 6.596 | 302.278 | 0.212 | 5.692 | 9.869 | 11.336 | 0.256 | 0.547 | 0.380 |
| RCAN | 4.709 | 272.189 | 0.200 | 3.062 | 9.772 | 12.055 | 0.227 | 0.558 | 0.325 |
| SRGAN-V | 10.007 | 291.546 | 0.210 | 35.932 | 8.276 | 10.448 | 0.286 | 0.477 | 0.557 |
| EDSR-V | 4.592 | 289.331 | 0.201 | 3.269 | 8.484 | 11.214 | 0.235 | 0.498 | 0.323 |
| ESRGAN-V | 7.187 | 413.398 | 0.213 | 4.010 | 7.887 | 10.695 | 0.309 | 0.469 | 0.399 |
| RBPN | 4.816 | 287.214 | 0.201 | 2.680 | 8.267 | 11.244 | 0.235 | 0.492 | 0.317 |
| EDVR | 2.148 | 213.034 | 0.179 | 1.352 | 8.479 | 10.060 | 0.180 | 0.476 | 0.329 |
| Ours | 4.198 | 221.859 | 0.191 | 1.890 | 7.723 | 9.568 | 0.197 | 0.441 | 0.312 |

Table 1: Cross-validation results. Comparison with state-of-the-art super resolution approaches. The best performance is marked with red (1st best), blue (2nd best). In practical applications, we recommend using only PEM, PDEM, and RMSE to evaluate model performance. For ease of use, we provide packages in our open source project that directly compute these metrics.

images (e.g., 3 frames, 5 frames.). In our experiments, we employ 5 frames. The core idea of VSR models is to increase the resolution by complementing texture information between different frames. It is worth mentioning that VSR models generally are equipped with a motion estimation module to alleviate the challenge of object motion to inter-frame information registration.

We evaluated 7 state-of-the-art SISR frameworks (i.e., Bicubic [29], SR-CNN [12], SRGAN [30], EDSR [33], ES-RGAN [58], DBPN [18], RCAN [67] and 5 VSR frameworks (i.e., SRGAN-V, EDSR-V, ESRGAN-V, RBPN [19], EDVR [57], of which 3 VSR methods (i.e., SRGAN-V, EDSR-V, ESRGAN-V) are modified from SISR. In order to verify the impact of increasing the number of input frames on performance, we build SRGAN-V, EDSR-V and ESRGAN-V by concatenating multiple frames of precipitation maps as the input of the model. In addition, we also evaluate Kriging [48], which is the de-facto standard method in the meteorological community. The mentioned 9 metrics are used to quantitatively evaluate the performance of these SR models and our method. Further, we select

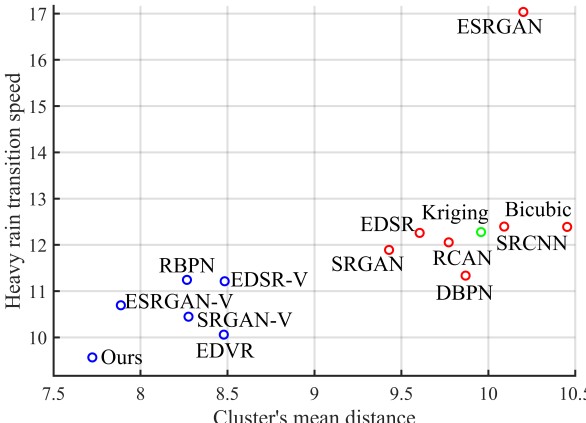

Figure 4: The dynamic property of benchmark algorithms. The frameworks of VSR are gathered in the lower-left corner of the figure, which demonstrates that VSR methods are superior to SISR and traditional methods in dynamic properties.

some disastrous weather as samples for qualitative analysis to test the model's ability to learn the dynamic properties of the weather system. And we employ the implementation of Pytorch for Bicubic. We use 4 NVIDIA 2080 Ti GPUs for training. We train all models with following setting. The batch size is set as 24. Precipitation maps are random crop into $64 \times 64$. We employ the Adam optimizer, beta1 is 0.9, and beta2 is 0.99. The initial learning rate is 0.001, which is reduced to 1/10 every 50 epochs, and a total of 200 epochs are trained. We evaluate benchmark frameworks with 17-fold cross-validation. The downscaling performances are shown in Tab. 1. We divide the indicators mentioned above into two groups. PDEM measures the model's ability to learn the dynamics of precipitation. PEM illustrates the model's ability to reconstruct precipitation.

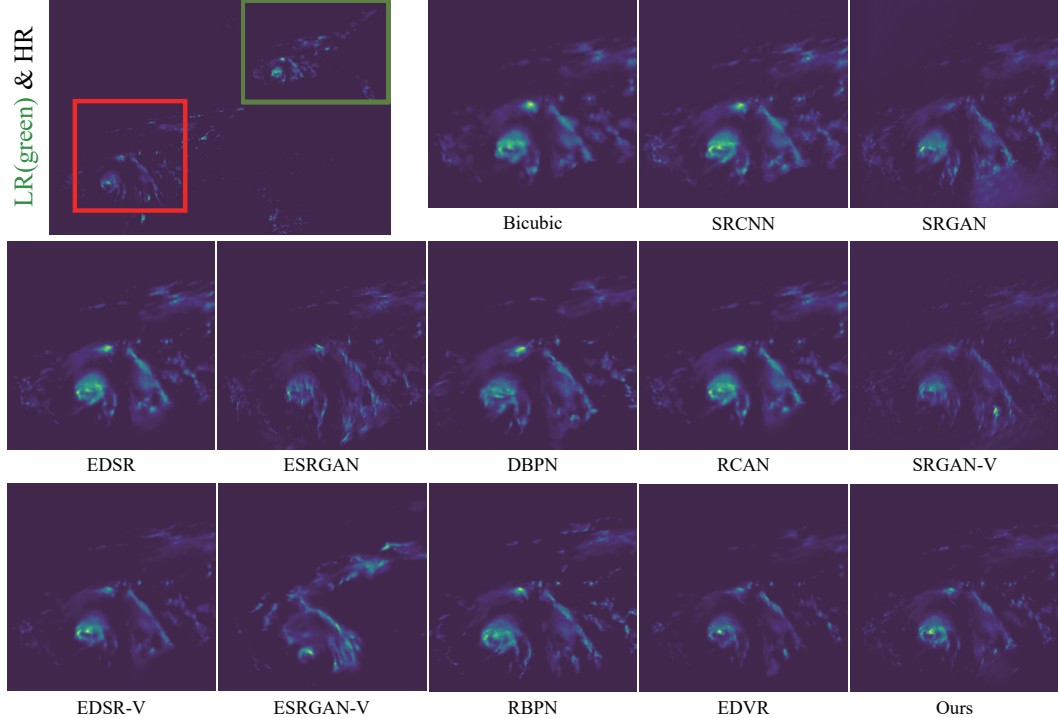

Figure 5: Visual comparison with state-of-the-art Super Resolution approaches. Please zoom in the figure for better observation. More results can be found in suppl. We show some video results in the suppl.

From Tab. 1, we can learn that the overall performance of the VSR methods are better than SISR models, which shows that the dynamic properties mentioned above are extremely important for the downscaling model. Furthermore, it can be seen from Fig. 4 that the SISR method is clustered in the upper right corner of the scatter plot, and the VSR method is concentrated in the lower-left corner, which further shows that the dynamic properties of the VSR methods are overall better than the SISR methods. In addition, our method achieves the $1st$ best performance in RMSE, PDEM, and achieve the second-best performance on PEM, which demonstrates that the proposed implicit dynamic estimation framework used is feasible and effective. It is worth mentioning that the traditional downscaling method Kriging performs better than many deep learning models (e.g., SRGAN, ESRGAN).

### 6.1.1 Qualitative analysis

We visualized the tropical cyclone precipitation map of the $166th$ hour (6th) in September 2010 and the high-resolution precipitation map generated by different methods. As shown in Fig. 5, the best perceptual effects are generated by EDVR and Our framework. Zooming in the result image, the precipitation maps generated by SRGAN and EDSR present obvious checkerboard artifacts. The reason for the checkerboard artifacts should be the relatively simple and sparse texture pattern in precipitation maps. The results generated by Bicubic, RCAN, Kriging, and SRCNN are over-smooth. DBPN even cannot reconstruct the eye of the hurricane. Especially, the result generated by Kriging is as fuzzy as the input LR precipitation map. In conclusion, the visual effects generated by the VSR methods are generally better than the SISR methods and the traditional method. From the perspective of quantitative and qualitative analysis, the dynamics estimation framework is very critical for downscaling.

## 7   Related Works

**Downscaling in Geoscience.** Downscaling is a fundamental task in geoscience and meteorology, which researchers have been interested in for a long time [60]. Most statistical downscaling methods regard this problem as point-wise regression [36, 53, 46] or direct maximum likelihood estimation of

the high-resolution data from the low-resolution data [2]. Bürger *et. al.* [5] compared bias-correction spatial disaggregation (BCSD), quantile regression networks, and expanded downscaling (XSD) for climate downscaling. Four fundamental statistical methods and three more advanced machine learning methods to downscale daily precipitation in the Northeast United States were compared in [53]. Recently, deep learning methods have been applied to solve the climate downscaling problem. DeepSD [54] tried to employ SRCNN [12] on precipitation downscaling. White *et. al.* [59] tested the performance of Generative Adversarial Networks (GANs) on downscaling weather models. ClimAlign [16] proposed a novel deep learning method for statistical downscaling, treating it as an unsupervised domain alignment problem without using paired low/high-resolution training data.

**Single Image Super-Resolution(SISR).** For deep learning based method, Dong *et. al.* first proposed an end-to-end model SRCNN [12] using deep convolutional network. SRGAN [30] proposed by Ledig *et. al.* uses perceptual loss for finer texture. The enhanced version ESRGAN [58] introduces RRDB as basic network unit, using relative discriminator and perceptual loss of features before activation.

**Video Super-Resolution(VSR).** VSR utilizes the temporal information of image sequences. Wang *et. al.* proposed a framework EDVR [57], which devises PCD module to handle large motion alignment, and the TSA fusion module with temporal and spatial attention. Xie *et. al.* proposed the tempoGAN [64] for fluid flow Super-Resolution, synthesizing 4-D physics fields with a temporal discriminator.

## 8 Future Work and Limitations

The recent success of vision based transformer (i.e., ViT) models [13, 34, 31, 7, 8, 42, 15] in image generation tasks has sparked a performance revolution. With its larger effective receptive field and high-order interaction, ViT greatly surpasses the previous dominant convolutional neural networks (i.e., CNNs) [20, 23, 24, 12, 30, 58, 57, 67, 18] in both video and image processing. Witnessing this situation, we believe that the introduction of the ViT model will be of great help to the real downscaling task presented by RainNet. At the same time, the diffusion based models [22, 9, 45, 41, 45] also shows unparalleled performance potential in generating images/videos (e.g., text to image), despite its notorious high computational complexity. In future work, we will deeply explore the tailored application of ViT model and diffusion-based model in RainNet to better solve this important meteorological problem.

## 9 Conclusion

In this paper, we built the first large-scale *real* precipitation downscaling dataset for the deep learning community. This dataset has 62424 pairs of HR and LR precipitation maps in total. We believe this dataset will further accelerate the research on precipitation downscaling. Furthermore, we analyze the problem in-depth and put forward three key challenges: temporal misalignment, temporal sparse and fluid properties. In addition, we propose an implicit dynamic estimation model to alleviate the above challenges. At the same time, we evaluated the mainstream SISR and VSR models and found that none of these models can perfectly solve RainNet's problems. Therefore, the downscaling task on this dataset is still very challenging. This work still remains several open problems. Currently, the data domain of this research is limited to the eastern U.S. In future research, we would enlarge the dataset to a larger domain. The dataset is only a single variable now. In future research, we may include more variables, e.g. temperature and wind speed.

### Acknowledgements

This work was supported by National Science Foundation of China (U20B2072, 61976137). We also appreciate the supporting of computing resources by the student innovation center of Shanghai Jiao Tong University.

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
