# OpenReview forum: "RainNet: A Large-Scale Imagery Dataset and Benchmark for Spatial Precipitation Downscaling"
_NeurIPS.cc/2022/Conference — NeurIPS 2022 Accept_

### Official Review · Reviewer_WVYT · 2022-07-09

**Rating:** 4
**Confidence:** 4
**Soundness:** 2 fair
**Presentation:** 2 fair
**Contribution:** 3 good

**Summary:**

This paper proposed a dataset consists of precipitation image sequences named SPDNet for spatial precipitation downscaling as well as a novel implicit dynamics estimation driven model. The proposed model as well as baseline models are evaluated on SPDNet with task specific metrics.

**Questions:**

1. What are the advantages of using the real data collected by two different systems for training super resolution models than using simple downsampling algorithms? If the two domains need to be similar, directly downsampling the high resolution data is both cheap and effective. If the two domains need to differ a lot from each other, it would be better to categorize the task as domain transferring instead of "spatial precipitation downscaling".
2. The qualitative results shown in Figure 4 are hard to distinguish. Could the authors provide more evidence to demonstrate that the models achieving better PEM and PDEM generate better predictions?

**Limitations:**

Listed in **Weaknesses**

**Strengths And Weaknesses:**

**Strengths**
1. **Valuable dataset**: The large scale dataset SPDNet is in high resolution and in sequence, which is a valuable contribution to data-driven meteorological research.
2. **Benchmark**: The authors have evaluate SOTA super resolution models on the proposed dataset ,and thus provide a good benchmark.

**Weaknesses**
1. **Necessity of low resolution data**: The authors claim in Introduction that the data obtained by simulated degradation is different from the real data collected by two different systems. However, there is no further discussion on it. It is insufficient to argue that their approach is better than obtaining low resolution data by simply downsampling the high resolution data. The author should compare these two approaches empirically, e.g., demonstrate that models trained with real data are able to reconstruct better high resolution data and hence boost the performance on downstream tasks.
2. **Unconvincing evaluation**: There are no clues about the effectiveness of proposed novel compound metrics PEM and PDEM. It would be much more convincing to conduct empirical studies to prove that models achieving better PEM/PDEM demonstrate better ability on addressing some concerned issues.
3. **Missing training details**: Selected baselines are not designed for precipitation data. It is necessary to (at least slightly) modify and tune the models for fair comparison. However, there is no information about these details except for a single statement "we also adjust the hype parameters of these models for better performance" in Supl. Sec.5.2.
4. **Formatting errors**: There are some language errors like "e.t.c." $\rightarrow$ "etc.", and misleading notations in mathematical expressions such as missing brackets in Eqn.1, missing description of "L", "H", "T" in line 250. Scores of 6 commonly used metrics in Table 1 should also be highlighted.

---

> ### Author Response · Authors · 2022-07-30
> **To Reviewer WVYT part 1**
>
> Thank you very much for your interest in our work and for your golden suggestions.
>
> Q1: What are the advantages of using the real data collected by two different systems for training super resolution models than using simple downsampling algorithms.
>
> A1: Thanks for the question. When considering real-world meteorological problems, the downscaling algorithm trained on data collected from two different systems will be more helpful. As we mentioned (lines 57-59), “Contrary to image data, the proposed real precipitation dataset … shows the physical characters (e.g., temporal misalignment, temporal sparse and fluid properties, etc., that challenge the downscaling algorithms.” The down-sampled dataset doesn’t reflect these real-world problems. It is necessary to emphasize that the difference between high-resolution observation data and low-resolution observation data in the real downscaling task [1] is not simply the difference in resolution, but the difference in observation methods (e.g., satellite and radar). This situation is like different degenerate kernels (e.g., unknown and bicubic) in image super-resolution. SR models trained on bicubic degenerate datasets (e.g., DIV2K-bicubic) suffer severe performance degradation on the in-the-wild raw data [2,3,4]. On the other hand, many parts of the world are covered by multiple-resolution observations of metrological variables. How to unify them and how to organize them become an important question. When it comes to the two systems mentioned in this dataset, NLDAS (lower-resolution) covers 1980-now, and StageIV (higher-resolution) covers 2002-now. Developing a downscaling algorithm to transfer NLDAS to StageIV allows researchers to extend higher-resolution observations of metrological variables to a longer period, which helps to understand the climate change effect on precipitation. We’ve added further explanations to the main text to explain the advantages of using the real data collected by two different systems.
>
> [1]. Reichstein M, Camps-Valls G, Stevens B, et al. Deep learning and process understanding for data-driven Earth system science[J]. Nature, 2019, 566(7743): 195-204.
>
> [2]. Ji, Xiaozhong, et al. "Real-world super-resolution via kernel estimation and noise injection." proceedings of the IEEE/CVF conference on computer vision and pattern recognition workshops. 2020.
>
> [3]. Hussein, et al. "Correction filter for single image super-resolution: Robustifying off-the-shelf deep super-resolvers." Proceedings of the IEEE/CVF Conference on Computer Vision and Pattern Recognition. 2020.
>
> [4]. Xu, Yu-Syuan, et al. "Unified dynamic convolutional network for super-resolution with variational degradations." Proceedings of the IEEE/CVF Conference on Computer Vision and Pattern Recognition. 2020.
>
> Q2: The author should compare these two approaches empirically, e.g., demonstrate that models trained with real data are able to reconstruct better high resolution data and hence boost the performance on downstream tasks.
>
> A2: Thank you for your constructive comments. We use the bicubic method (Widely used to synthesize data) to downsample the high-resolution data (624*999) from 2002.7 to 2016.11 to low-resolution data (208 × 333), so that we generate a synthetic dataset. We employ this dataset to train our model from scratch, and use the original data from 2017.7\~2017.11 as the test set. We report the test results in the table below:
>
> |  Approach   | MPPE &darr;  | HRRE &darr;  | AMMD &darr;  | CPMSE &darr;  | HRTS &darr;  | CMD &darr; | PEM &darr; | PDEM &darr; | RMSE X100 &darr; |
> |  ----  | ----  | ----  | ----  |----  | ----  | ----  | ----  | ----  | ----  |
> | Ours (real data)  | 4.198 | 221.859 | 0.191 | 1.890 | 7.723 | 9.568 | 0.197 | 0.441 | 0.312 |
> | Ours (synthetic data)  | 5.187 | 311.212 | 0.232 | 3.121 | 9.953 | 12.282 | 0.259 | 0.568 | 0.399 |
>
> It can be seen from the table that the performance of the model trained on the bicubic synthetic dataset (row #3) is severely degraded. Therefore, the model trained with the real collected data has a great advantage in the task of spatial precipitation downscaling, and also confirms "A1". We have added the above experiment to the Sec.6.1 of the new version of our paper.

---

> > ### Comment · Reviewer_WVYT · 2022-08-09
> > **Thanks for your response**
> >
> > I'm sorry for the late response. The authors addressed some of my questions given the limited time. Thanks.
> >
> > In the response to Q1, the authors also provide detailed explanation on the necessity of real low-res data and empirical study on the corresponding improvement. The response seems reasonable to me.
> >
> > As for Q3 and Q6, the evaluation of performance on precipitation related tasks is still an open problem. E.g., DeepMind's Nature paper resorted to meteorologists for human evaluations due to the discrepancy between evaluations from experts and scores. It's not appropriate to include the intuitive designs of PEM/PDEM as one of the major contributions in this paper.
> >
> > Overall, the dataset and the corresponding benchmark are valuable. I suggest that the authors focus on them and remove the PEM/PDEM part in the paper.  The value of the dataset and benchmark will not be diminished by not proposing "novel" metrics. The proposed method does not necessarily have to outperform baselines in all concerned metrics.
> >
> > [1] Ravuri, Suman, et al. "Skilfull precipitation nowcasting using deep generative models of radar." Nature 597.7878 (2021): 672-677.

---

> > > ### Author Response · Authors · 2022-08-09
> > > **To Reviewer WVYT**
> > >
> > > We sincerely thank you for the review and comments.
> > >
> > > Also thank you for acknowledging the value of our dataset.
> > >
> > > After discussions in our team, we thought that we should reduce the discussion of metrics and focus on metric that are very familiar to the computer field (such as RMSE).
> > > We will add more content to introduce the dataset and benchmark itself so that this paper focuses on our propsed dataset and model.
> > > More dataset-related details, as well as model design and training details, will be covered in the main text.
> > >
> > > Best, Authors of Paper 690

---

> ### Author Response · Authors · 2022-08-01
> **To Reviewer WVYT part2**
>
> Q3: There are no clues about the effectiveness of proposed novel compound metrics PEM and PDEM. It would be much more convincing to conduct empirical studies to prove that models achieving better PEM/PDEM demonstrate better ability on addressing some concerned issues.
>
> A3: Thanks for the comment. Here we are trying to make the metrics more applicable to meteorology society while also containing variables (e.g., RMSE) that are familiar to the computer science society. The metrics Precipitation Error Measure (PEM) and Precipitation Dynamics Error Measure (PDEM) are weighted over a series of metrics with clear physical meaning (reconstruction metrics: MPPE, HRRE, CPMSE, AMMD, and dynamic metrics: HRTS and CMD) and have been applied in downscaling research in meteorology society for a long time (may not in the same abbreviation). We’ve mentioned these in the supplementary (section 3. Metrics) and have added an explanation to the main text. In supplementary section 3, we also discussed how each metric is calculated and what other literature employs the metric. This explains why better PEM/PDEM demonstrates a better ability to address downscaling problems (from a meteorology sense). For example (line 30-33 in supplementary), “The mesoscale peak precipitation error (MPPE; mm/hour) is calculated as the difference of top quantile between the generated/real rainfall dataset which considering both spatial and temporal property of mesoscale meteorological systems, e.g., hurricane, squall. This metric is used in most of these papers (for example [15, 10, 2, 6, 11, 14] (refs in our paper) suggest the quantile analysis to evaluate the downscaling quality)”. To be noticed, in meteorology society, researchers tend to evaluate one downscaling algorithm with not a single variable but multiple variables together. However, it is always important to condense the information when bridging two fields. Here we weighted these variables into two to make comparing machine learning models easier for computer science society.
>
> Q4: Missing training details: Selected baselines are not designed for precipitation data. It is necessary to (at least slightly) modify and tune the models for fair comparison. However, there is no information about these details except for a single statement "we also adjust the hype parameters of these models for better performance" in Supl. Sec.5.2.
>
> A4:  Thanks for the question. The parts that need to be adjusted for these models include two parts：
> 1. The hyperparameters required for training, which we have explained in line 309\~313 of our paper. It is worth pointing out that typically SR models use a learning rate of 1e-4\~5e-4, but we found that using 1e-3 is better for our task.
> We have added more detailed instructions in Section 6.1 of the new version of our paper;
> 2. The adjustment of the model, including the adjustment of the input channel and upsampling rate. We adjust the number of channels of input data for SRCNN, SRGAN, EDSR, ESRGAN, DBPN, RCAN, EDVR, RBPN, and our model to 1. We set the input data channel to 5 for SRGAN-V, EDSR-V, and ESRGAN-V. We set the upsampling rate to 3 for all models.
>
>
> Q5: Formatting errors.
>
> A5:  Thank you for your thoughtful suggestions. We had corrected "e.t.c" to "etc.". Careful corrections have been made to the language of our paper. "L" denots low-resolution and "H" denots the high-resolution. "T" represents the frame number. A detailed explanation has been added to Section 5 of the new version of our paper. We have revised the writing of Eq.1. We had highlighted 6 commonly used metrics in Table 1 of the new version of our paper.
>
> Q6: The qualitative results shown in Figure 4 are hard to distinguish. Could the authors provide more evidence to demonstrate that the models achieving better PEM and PDEM generate better predictions?
>
> A6: Thank you for your thoughtful suggestion. For this task, visualization is only an auxiliary means. The field of meteorology directly uses the quantitative metrics mentioned in our paper to measure the quality of model predictions, as described in A3. The two metrics PEM and PDEM describe overall performance over a period of time, while PDEM describes dynamic properties that are difficult to capture in static pictures. PEM and RMSE are usually positively correlated. More reflected in the visualization is the level of RMSE. It is worth mentioning that PSNR/SSIM/LPIPS visual effects are often indistinguishable in image super-resolution tasks. To improve the distinguishability of qualitative analysis results, we marked the PEM and RMSE corresponding to the visualization results to facilitate readers to distinguish. Furthermore, the discriminative regions in the visualization are marked with red boxes.

---

> ### Author Response · Authors · 2022-08-09
> **To reviewer WVYT**
>
> Dear reviewer WVYT:
>
> We sincerely thank you for the review and comments. We have provided corresponding responses and results, which we believe have covered your concerns. We hope to further discuss with you whether or not your concerns have been addressed. Please let us know if you still have any unclear parts of our work.
>
> Best, Authors of Paper 690

---

### Official Review · Reviewer_aP2Q · 2022-07-11

**Rating:** 6
**Confidence:** 4
**Soundness:** 3 good
**Presentation:** 4 excellent
**Contribution:** 3 good

**Summary:**

The paper proposed a large-scale spatial precipitation downscaling dataset named SPDNet. The dataset contains more than 62400 pairs of high-quality low/high-resolution precipitation maps for over 17 years, and covers more than 9 million squre kilometers of land area. The author also introduced 6 metrics that evaluate different aspects of the downscaling models, and 2 summary metrics that combine these 6 individual metrics. The author viewed the task as a video super-resolution problem and compared 14 methods. From the experimental results, the overall performance of the video super-resolution (VSR) models are better than Single-Image Super-Resolution (SISR) models.


**Questions:**

1. The author mentioned that the dataset contains lots of different events such as hurricane, squall. Are the sequences in the dataset marked with the event type?


**Limitations:**

1. The paper will be limited regarding the coverage of the baseline methods. However, it is difficult to cover all the latest image super-resolution methods so it is acceptable.



**Strengths And Weaknesses:**

Stengths:

1. The paper proposed the first large-scale precipitation downscaling dataset. This is an important scientific problem and a large-scale dataset can help move the area forward. In addition, the author pointed out the unique characteristics of the task such as temporal misalignment, temporal sparse, and fluid properties.
2. The paper proposed 6 metrics for evaluating the models, including 4 reconstruction metrics that focuses on evaluating if the predicted high-resolution precipitation map matches the ground-truth, and 2 dynamic metrics that evaluate the dynamics of the predicted precipitation (via first order dynamics).
3. The paper compared 14 models, including the Kriging method that has been widely used in the geospatial community, and other SISR and VSR methods. The author also extended SRGAN, EDSR, ESRGAN to be VSR methods.

Weaknesses
1. It seems that RMSE is itself a very good summary metric. Thus, it is not clear why we will still need PEM / PDEM.
2. Currently, the state-of-the-art image super-resolution model is based on vision Transformers (e.g., SwinIR) and the author need to reference the latest progress in this area.

---

> ### Author Response · Authors · 2022-07-30
> **To Reviewer aP2Q**
>
> Thank you very much for your interest in our work and for your valuable comments.  This would be an important work bridging meteorology and computer science. In this paper, we propose the first large-scale dataset for precipitation downscaling that is based on real measured data while the previous models are usually evaluated on synthetic datasets (downsampling the radar maps to generate the synthetic low/high-resolution pairs) and no formal dataset released previously. Under the general trend of the times, it is always good to extend from AI to AI+X. Alphafold's success is such a good example, which tells that deep and well-communicated interaction between AI and other fields could stimulate large scientific breakthroughs. Downscaling is one of the most important tasks in current meteorological research, and the combination with deep learning is also the main research trend [a]. We believe this paper is also a meaningful and successful one and time proves it. To accomplish this work, great and difficult communications between computer science and meteorology side have been done to ensure this precipitation down-scaling is the most important and cutting-edge meteorological task that could be handle by computer science.
>
> [a]. Reichstein M, Camps-Valls G, Stevens B, et al. Deep learning and process understanding for data-driven Earth system science[J]. Nature, 2019, 566(7743): 195-204.
>
> Q1: It seems that RMSE is itself a very good summary metric. Thus, it is not clear why we will still need PEM / PDEM.
>
> A1: Thanks for the comment. We agree RMSE is an excellent summary metric and which is also very familiar to computer science society. For meteorology society, researchers usually use metrics that consider many kinds of meteorological phenomena. The metrics Precipitation Error Measure (PEM) and Precipitation Dynamics Error Measure (PDEM) are weighted over a series of metrics with clear physical meaning (reconstruction metrics: MPPE, HRRE, CPMSE, AMMD, and dynamic metrics: HRTS and CMD) and have been applied in downscaling research in meteorology society for a long time (may not in the same abbreviation). We could expect better PEM/PDEM performance when better RMSE performance is observed, but it might not always be the case. To make this dataset practical for computer scientists and meteorologists, here we introduce both PEM/PDEM and RMSE systems to benchmark the algorithms.
> For details on calculating PEM/PDEM, we’ve mentioned these in the supplementary (Section 3. Metrics) and have added an explanation to the main text. In supplementary Section 3, we also discussed how each metric is calculated and what other literature employs the metric. This explains why better PEM/PDEM demonstrates a better ability to address downscaling problems (from a meteorology sense). For example (line 30-33 in supplementary), “The mesoscale peak precipitation error (MPPE; mm/hour) is calculated as the difference of top quantile between the generated/real rainfall dataset which considering both spatial and temporal property of mesoscale meteorological systems, e.g., hurricane, squall. This metric is used in most of these papers (for example [15, 10, 2, 6, 11, 14] (refs in our paper) suggest the quantile analysis to evaluate the downscaling quality)”. To be noticed, in meteorology society, researchers tend to evaluate one downscaling algorithm with not a single variable but multiple variables together. However, it is always important to condense the information when bridging two fields. Here we weighted these variables to two to make it easier to compare machine learning models and easier for computer science society to follow.
>
>
> Q2: Currently, the state-of-the-art image super-resolution model is based on vision Transformers (e.g., SwinIR) and the author need to reference the latest progress in this area.
>
> A2: Thank you for your constructive comments. We will definitely add the state-of-the-art transformer-based SR models (e.g., SwinIR) trained on our dataset to the benchmark models.
>
> Q3: The author mentioned that the dataset contains lots of different events such as hurricane, squall. Are the sequences in the dataset marked with the event type?
>
> A3: Thanks for the comment. Yes, we have provided event annotations such as hurricanes, squall lines for relevant frames.

---

> > ### Comment · Reviewer_aP2Q · 2022-08-08
> > **Thanks for the rebuttal**
> >
> > Thanks for the rebuttal.
> >
> > Regarding Q1, I can understand the necessity for including a few domain-specific score functions such as  MPPE, HRRE, CPMSE, AMMD, HRTS, CMD. However, it is not clear why we need PEM / PDEM at this stage given that they are very consistent with the simpler MSE score. For example, there might be other domain-specific scores that are missing in the benchmark, how should we incorporate these scores in the PEM / PDEM? In fact, people may later adopt these metrics for model selection. Can we get a model that is better at domain specific scores by selecting the candidate models with PEM/PDEM and not MSE?
> >
> > Regarding Q2, thanks for agreeing to add vision transformers in the benchmark.
> >
> > Regarding Q3, thanks for the reply. The event types can be useful for further analyzing whether the SR algorithms are robust for different domains (i.e., meteorological events).

---

> > > ### Author Response · Authors · 2022-08-08
> > > **To reviewer aP2Q**
> > >
> > > Thanks for your valuable reply!
> > >
> > > Q1: Can we get a model that is better at domain specific scores by selecting the candidate models with PEM/PDEM and not MSE?
> > >
> > > A1: Thank you for your question. In spatial precipitation downscaling tasks, domain researchers typically use the metrics introduced in our paper to evaluate/select models instead of directly using RMSE [1]. The RMSE here is the pixel-level average error over all frames (i.e. 720 frames in a month), this averaging loses structural and dynamic information [1], which are properties of most interest to domain researchers. "We could expect better PEM/PDEM performance when better RMSE performance is observed, but it might not always be the case."  For example, the model can reconstruct some frames very well and others very poorly (this often happens in heavy rain situations, e.g., hurricanes, continuous heavy rain, they occur almost every year), and the model can also get decent RMSE values, in this case, the model exhibits poor temporal consistency and dynamics. However, these issues can be captured by PDE/PDEM, which are more fine-grained metrics.
> > > In other words, similar RMSEs may have different PDEs and PDEMs, for example, RCAN (RMSE\times 100:0.325, PEM: 0.227, PDEM: 0.558) and EDVR (RMSE X100:0.329, PEM: 0.180, PDEM: 0.476) ). So simply using RMSE may cause models selection to fail.
> > > In fact, CPMSE has similar functionality to RMSE.
> > > Therefore, it is entirely feasible to use PDE/PDEM directly for model selection or evaluation.
> > >
> > > [1]. Ekström, Marie. "Metrics to identify meaningful downscaling skill in WRF simulations of intense rainfall events." Environmental Modelling & Software 79 (2016): 267-284.
> > >
> > > Q2: For example, there might be other domain-specific scores that are missing in the benchmark, how should we incorporate these scores in the PEM / PDEM?
> > >
> > > A2: Thank you for your question. It can be added to PEM/PDEM by first normalizing and then weighting the metric to be added.

---

### Official Review · Reviewer_5jsP · 2022-07-11

**Rating:** 6
**Confidence:** 2
**Soundness:** 2 fair
**Presentation:** 4 excellent
**Contribution:** 3 good

**Summary:**

This paper presents a large scale dataset for spatial precipitation downscaling which contains more than 62, 400 pairs of high-quality low/high-resolution precipitation maps for over 17 years, ready to help the evolution of deep learning models in precipitation
downscaling. The precipitation maps  collected in the dataset  cover various meteorological phenomena such as hurricane and  squall. The data are organised in time series of maps, with 720 maps/month. Comprehensive metrics are also provided to evaluate the performances of models.

**Questions:**

Eastern coast of US has been selected for data collections. What about other regions ?

**Ethics Review Area:**

["I don’t know"]

**Limitations:**

The dataset lacks precipitation maps for several regions

**Strengths And Weaknesses:**

This paper is well written and brings comprehensive dataset for spatial precipitation. However  the technical novelty is low.

---

> ### Author Response · Authors · 2022-07-30
> **To Reviewer 5jsP**
>
> Thank you very much for your interest in our work and for your valuable comments. This would be an important work bridging meteorology and computer science. In this paper, we propose the ***first*** large-scale dataset for precipitation downscaling that is based on real measured data while the previous models are usually evaluated on synthetic datasets (downsampling the radar maps to generate the synthetic low/high-resolution pairs) and no formal dataset released previously. Under the general trend of the times, it is always good to extend from AI to AI+X. Alphafold's success is such a good example, which tells that deep and well-communicated interaction between AI and other fields could stimulate large scientific breakthroughs. Downscaling is one of the most important tasks in current meteorological research, and the combination with deep learning is also the main research trend [1]. We believe this paper is also a meaningful and successful one and time proves it. To accomplish this work, great and difficult communications between computer science and meteorology side have been done to ensure this precipitation down-scaling is the most important and cutting-edge meteorological task that could be handle by computer science.
>
> [1]. Reichstein M, Camps-Valls G, Stevens B, et al. Deep learning and process understanding for data-driven Earth system science[J]. Nature, 2019, 566(7743): 195-204.
>
> For the technical novelty concern.
>
> Our work demonstrates its novelty in two aspects:
>
> 1.The first large-scale open-source dataset for precipitation downscaling that is based on real measured data as described above, which will greatly help bridge the DL/ML community with meteorological science, while promoting the development of AI-for-Science.
>
> 2.Novel benchmark model structure design and performance. Existing VSR methods generally include motion estimation modules, which are composed of modules (e.g., PCD in EDVR, Projection Module in RBPN, etc.) with strong video dynamics assumptions. As mentioned in our paper, the assumptions do not match precipitation downscaling. Unlike them, our implicit dynamic estimation module (IDEM) is a low inductive-bias module (e.g., transformers outperform CNNs), it only contains N-2 (N is the input adjacent frames, 5 frames in our model setting) weight-sharing small networks, so that IDEM can explore the inherent laws in the precipitation data without constraints/assumptions. In addition, self-attention, as a low inductive-bias operator, has achieved huge performance improvements in computer vision tasks (e.g., image classification, object detection, etc.). The low inductive-bias setting allows self-attention to fully explore the inherent laws within the data without being constrained by data assumptions [2]. At the same time, the self-attention operator also exhibits stronger generalization ability. Analogously, this is also the potential reason why our IDEM works better on the precipitation dataset. The results in Table 1 (in our paper) show the superiority of our model. Furthermore, our IDEM module also shows very competitive performance on the VSR data set: Vid4(4×) Average RGB PSNR 25.85 (EDVR Average RGB PSNR 25.83, DUF Average RGB PSNR 25.79). We will add more details (novelty analysis and performance analysis in VSR task) about the proposed model in Sec.5 and Sec.6.
>
> [2]. Esser P, Rombach R, Ommer B. Taming transformers for high-resolution image synthesis[C]//Proceedings of the IEEE/CVF Conference on Computer Vision and Pattern Recognition. 2021: 12873-12883.
>
> Q1: Eastern coast of US has been selected for data collections. What about other regions ?
>
> A1: Thank you for your constructive comments. There several reasons for selecting the eastern coast of US.
> 1. Compared with other regions in the world, the US has systematic and complete observational data (NLDAS (lower-resolution) covers 1980-now, and StageIV (higher-resolution) covers 2002-now) of various resolutions from different observational systems (e.g., satellite, weather radar, etc.).
> 2. Compared to the eastern US, the West Coast has very little precipitation, which is not helpful for our task, so we discarded the West Coast to reduce the redundancy of the dataset.
> 3. In our future work, we will expand to more regions of the world.

---

> ### Author Response · Authors · 2022-08-09
> **To reviewer 5jsP**
>
> Dear reviewer 5jsP:
>
> We sincerely thank you for the review and comments. We have provided corresponding responses, which we believe have covered your concerns. We hope to further discuss with you whether or not your concerns have been addressed. Please let us know if you still have any unclear parts of our work.
>
> Best, Authors of Paper 690

---

### Official Review · Reviewer_9mB6 · 2022-07-26

**Rating:** 5
**Confidence:** 3
**Soundness:** 3 good
**Presentation:** 3 good
**Contribution:** 3 good

**Summary:**

There is not much to say here. The paper organizes a large precipitation dataset from both high res and low res sources and illustrates some baselines. The problem addressed is both interesting to ML (esp. to those interested in mixing physics and ml), and important. The question then, is; given that this data can be obtained directly from the original sources, is a new organization of it necessary and does it warrant a publication in an ML conference. Although I expect other reviewers will disagree, I have a positive view. Publishing this paper will probably increase interest in ML in this set of problems, and some in the community will find the dataset useful.

**Questions:**

SPDnet is not a very good name…

**Limitations:**

It says in the checklist that the code and the data are proprietary. This needs to be clarified. What’s the point of publishing a dataset if it cannot be used by others?

**Strengths And Weaknesses:**

See above

---

> ### Author Response · Authors · 2022-07-30
> **To Reviewer 9mB6**
>
> Thank you very much for your interest in our work and for your golden suggestions. This would be an important work bridging meteorology and computer science. In this paper, we propose the first large-scale dataset for precipitation downscaling that is based on real measured data while the previous models are usually evaluated on synthetic datasets (downsampling the radar maps to generate the synthetic low/high-resolution pairs) and no formal dataset released previously. Under the general trend of the times, it is always good to extend from AI to AI+X. Alphafold's success is such a good example, which tells that deep and well-communicated interaction between AI and other fields could stimulate large scientific breakthroughs. Downscaling is one of the most important tasks in current meteorological research, and the combination with deep learning is also the main research trend [1]. We believe this paper is also a meaningful and successful one and time proves it. To accomplish this work, great and difficult communications between computer science and meteorology side have been done to ensure this precipitation down-scaling is the most important and cutting-edge meteorological task that could be handle by computer science.
>
> [1]. Reichstein M, Camps-Valls G, Stevens B, et al. Deep learning and process understanding for data-driven Earth system science[J]. Nature, 2019, 566(7743): 195-204.
>
> Q1: SPDnet is not a very good name…
>
> A1: Thank you for your constructive comments. SPDNet is a straightforward name derived from shorthand for "Spatial Precipitation Downscaling".
> We believe a good name is very important to our work, so we will look for a better one.
>
> Q2:  "It says in the checklist that the code and the data are proprietary. This needs to be clarified. What’s the point of publishing a dataset if it cannot be used by others?"
>
> A2:
> Thanks for the comment.
> ***All relevant codes and datasets are open-source for research purposes.***
> We apologize for the error in filling out the checklist, we have changed the item "Do you include licenses for code and datasets? [No] Code and data are proprietary" to "Do you include licenses for code and datasets?[ Yes] see Section 4.2".
> The dataset and the code are not proprietary.
> We will build a dedicated github repository and website for users to easily use our datasets and codes.

---

> ### Author Response · Authors · 2022-08-09
> **To reviewer 9mB6**
>
> Dear reviewer 9mB6:
>
> We sincerely thank you for the review and comments. We have provided corresponding responses, which we believe have covered your concerns. We hope to further discuss with you whether or not your concerns have been addressed. Please let us know if you still have any unclear parts of our work.
>
> Best,
> Authors of Paper 690

---

### Meta-Review · Area_Chair_KSxd · 2022-08-26

**Recommendation:** Accept
**Confidence:** Less certain

**Metareview:**

This paper describes SPDNet, a dataset for spatial precipitation downscaling.

Experiments are provided using a fairly wide set of alternative methods - 14 models (including Kriging which is a widely used standard method in the meteorological community) - as well as a novel architecture proposed by the authors. The authors also extended SRGAN, EDSR, ESRGAN from Single Image Super Resolution (SISR) methods to Video Super Resolution (VSR) methods. While the level of innovation on the neural architecture side of the work is not extreme, clear value is provided in terms of contributions to neural architecture development. Reviewers felt that the dataset itself, the wide variety of models examined and the large set of evaluation metrics offers value to the community and that this dataset could help bring more interest to the problem domain.

During the discussion period it was made clear that "All relevant codes and datasets are open-source for research purposes" and that
"The dataset and the code are not proprietary. We will build a dedicated github repository and website for users to easily use our datasets and codes." It is important that this is indeed is fully executed by the authors.

Three of four reviewers recommended acceptance.

For all these reasons the AC recommends acceptance.

**Award:**

No

---

### Decision · Program_Chairs · 2022-09-14

Accept